# Chromatin and gene expression changes during female *Drosophila* germline stem cell development illuminate the biology of highly potent stem cells

**Liang-Yu Pang, Steven DeLuca†, Haolong Zhu, John M Urban, Allan C Spradling***

Howard Hughes Medical Institute, Carnegie Institution for Science, Baltimore, United States

**Abstract** Highly potent animal stem cells either self renew or launch complex differentiation programs, using mechanisms that are only partly understood. *Drosophila* female germline stem cells (GSCs) perpetuate without change over evolutionary time and generate cystoblast daughters that develop into nurse cells and oocytes. Cystoblasts initiate differentiation by generating a transient syncytial state, the germline cyst, and by increasing pericentromeric H3K9me3 modification, actions likely to suppress transposable element activity. Relatively open GSC chromatin is further restricted by Polycomb repression of testis or somatic cell-expressed genes briefly active in early female germ cells. Subsequently, Neijre/CBP and Myc help upregulate growth and reprogram GSC metabolism by altering mitochondrial transmembrane transport, gluconeogenesis, and other processes. In all these respects GSC differentiation resembles development of the totipotent zygote. We propose that the totipotent stem cell state was shaped by the need to resist transposon activity over evolutionary timescales.

## eLife assessment

This **important** work significantly advances our comprehension of the molecular events occurring during germline stem cell differentiation in the *Drosophila melanogaster* ovary. The conclusions are strongly supported by **compelling** evidence, including rigorous data sets and complementary whole-genome analyses. As a result, this research holds substantial interest for developmental and stem cell biologists alike.

## Introduction

The animal zygote launches embryonic development from a totipotent chromatin state in a process that has been intensively studied using model invertebrate and vertebrate organisms (reviewed in *Macrae et al., 2023*; *Zhou and Cho, 2022*; *Ahmad and Henikoff, 2022*; *Gleason and Chen, 2023*). Pluripotent cell lines derived from embryos (*Thomson et al., 1998*) or reprogramed somatic cells (*Takahashi and Yamanaka, 2006*) provide technically favorable material for analyzing the gene expression, chromatin, and metabolic states that drive stem cells to diverse differentiated derivatives (*Rafalski et al., 2012*; *Kinoshita and Smith, 2018*; *Omole and Fakoya, 2018*; *Soldner and Jaenisch, 2018*). Highly potent stem cells in vivo and in cell lines cycle in an unusual manner with very short G1 phases, express genes in a variable manner, and show 'bivalent' chromatin marks (*Hsu et al., 2008*; *Pauklin and Vallier, 2013*; *Ter Huurne and Stunnenberg, 2021*). During subsequent development, daughter cells drastically alter their chromatin, gene expression, organelles, and metabolism. These

**\*For correspondence:** spradling@carnegiescience.edu

**Present address:** †Montana State University, Bozeman, United States

**Competing interest:** The authors declare that no competing interests exist.

**Preprint posted** 25 June 2023

**Sent for Review** 05 July 2023

**Reviewed preprint posted** 31 August 2023

**Version of Record published** 13 October 2023

**eLife digest** Most animals are made up of two cell types: germline stem cells, which give rise to reproductive cells (egg and sperm) and pass their DNA to the next generation, and somatic cells, which make up the rest of the body.

Transposable elements – fragments of DNA that can copy themselves and integrate into different parts of the genome – can greatly disrupt the integrity of the germ cell genome. Systems involving small RNAs and DNA methylation, which respectively modify the sequence and structure of the genome, can protect germ cells from the activity of transposable elements. While these systems have been studied extensively in late germ cells, less is known about how they work in germ cells generated early on in development.

To investigate, Pang et al. studied the germline stem cells that give rise to eggs in female fruit flies. Techniques that measure DNA modifications showed that these germline stem cells and the cells they give rise to early on are better protected against transposable elements. This is likely due to the unusual cell cycle of early germ cells, which display a very short initial growth phase and special DNA replication timing during the synthesis phase. Until now, the purpose of these long-known cell cycle differences between early and late germ cells was not understood.

Experiments also showed known transposable element defences are upregulated before the cell division that produces reproductive cells. DNA becomes more densely packed and germ cells connect with one another, forming germline 'cysts' that allow them to share small RNAs that can suppress transposable elements. Pang et al. propose that these changes compensate for the loss of enhanced repression that occurs in the earlier stem cell stage. Very similar changes also take place in the cells generated from fertilized eggs and in mammalian reproductive cells.

Further experiments investigated how these changes impact the transition from stem cell to egg cell, revealing that germline stem cells express a wide diversity of genes, including most genes whose transcripts will be stored in the mature egg later on. Another type of cell produced by germline stem cells known as nurse cells, which synthesize most of the contents of the egg, dramatically upregulate genes supporting growth. Meanwhile, 25% of genes initially expressed in germline stem cells are switched off during the transition, partly due to a mechanism called Polycomb-mediated repression.

The findings advance fundamental knowledge of how germline stem cells become egg cells, and could lead to important findings in developmental biology. Furthermore, understanding that for practical applications germline stem cells do not need to retain transposable element controls designed for evolutionary time scales means that removing them may make it easier to obtain and manipulate new stem cell lines and to develop new medical therapies.

changes parallel reduced developmental potency but the evolutionary forces that gave rise to animal stem cells remain little understood.

*Drosophila* female germline stem cells provide an attractive system for studying totipotency maintenance and differentiation (reviews: *Fuller and Spradling, 2007*; *Drummond-Barbosa, 2019*; *Hinnant et al., 2020*). New female germ cells continuously arise downstream from germline stem cells (GSCs) located within the adult germarium, and develop into ovarian follicles ordered by age within ovarioles (*Figure 1A*). GSC daughters initially divide synchronously to build 16-cell cysts that specify an oocyte and 15 nurse cells (NCs), traverse meiotic prophase, become associated with somatic follicle cells (FCs) and leave the germarium as stage 2 (S2) follicles. Over the next 3 days, follicle development increases the oocyte volume to more than 10,000 times that of a GSC (*Figure 1—figure supplement 1*, *Figure 1—figure supplement 2*). Following fertilization of the mature S14 oocyte, the resulting totipotent zygote develops as a syncytial embryo before germ cells bud from the posterior pole, migrate to the gonad and complete the generational cycle by forming new GSCs (*Figure 1—figure supplement 1*). Previous studies (reviewed in *Matova and Cooley, 2001*; *Lu et al., 2017*; *Hinnant et al., 2020*; *Spradling et al., 2022*) show that aspects of oocyte development have been conserved between *Drosophila*, mouse and other animals, including germline cyst and NC formation (*Lei and Spradling, 2016*; *Niu and Spradling, 2022*), NC turnover (*Lebo and McCall, 2021*; *Niu and Spradling, 2022*), and Balbiani body production (*Mahowald and Strassheim, 1970*; *Pepling et al., 2007*).

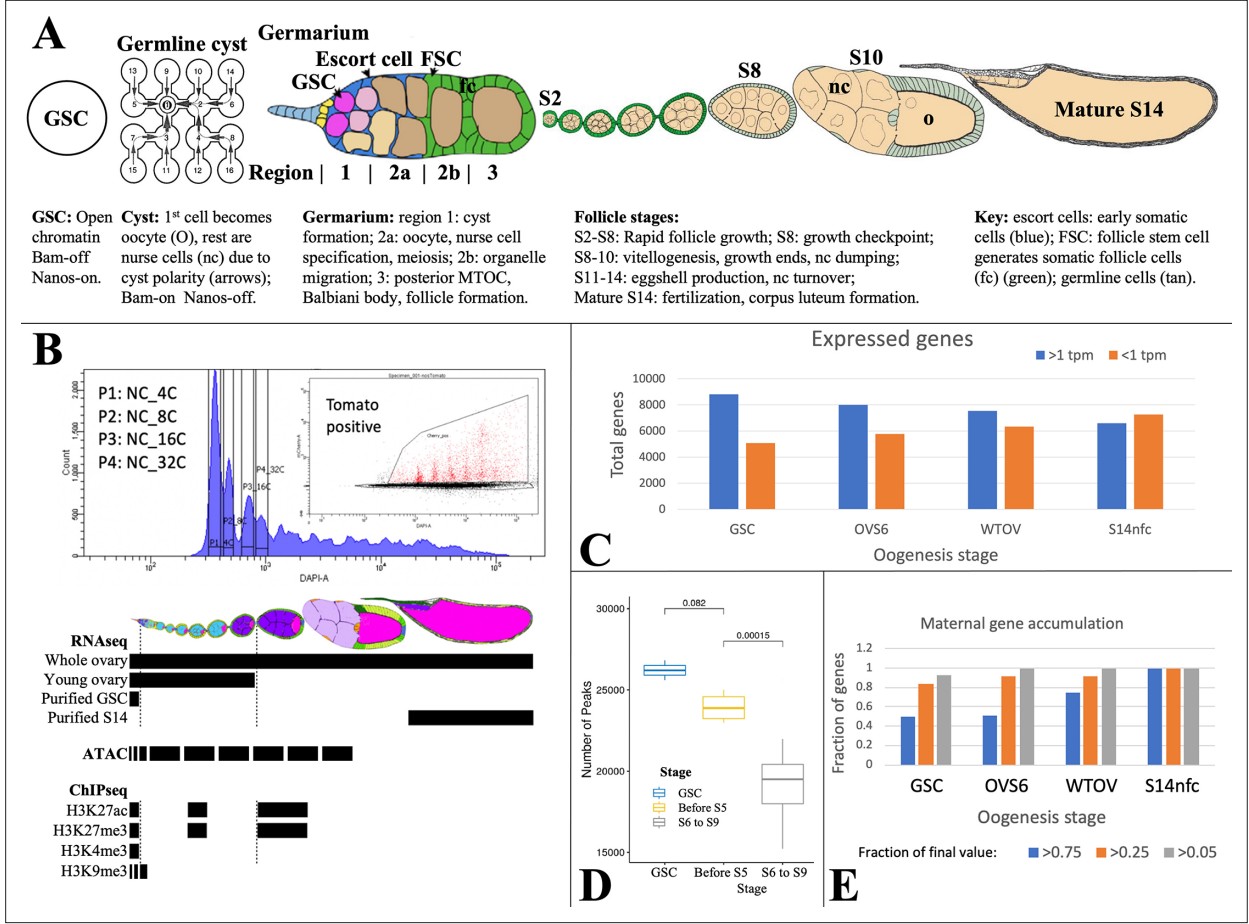

**Figure 1.** Analyzing *Drosophila* female germline chromatin. (**A**) Stages of female gamete development in an ovariole. A single germline stem cell (GSC) and germline cyst are shown. To the right, a germarium is illustrated showing regions 1–3. Follicles in an ovariole are pictured at a lower magnification starting with stage 2 (S2). Below the diagrams, major events are summarized. (**B**) Fluorescence-activated cell sorting (FACS) purification (upper panel) of 4C-512C germ cells for ATAC after earlier separation of 2C and 4C GSCs, and 2C-16C follicle cells (see *Figure 2*). DAPI, DNA content; inset shows further purification based on germ cell marker expression (tomato). Below, follicle stages collected for RNA-seq and Chip-seq are shown by black bars. (**C**). The number of genes expressed (>1 tpm, blue) and off (<1 tpm, orange) in GSCs decreases as germ cell development proceeds downstream. (**D**) Box plot showing a decline in total ATAC peaks from GSCs to young ovaries (before S6) to whole ovaries (S6-9). (**E**) Stem cells already express a high fraction of 79 oocyte maternal effect genes defined in *Drosophila* genetic screens.

The online version of this article includes the following figure supplement(s) for figure 1:

**Figure supplement 1.** The *Drosophila* female germ cell cycle.

**Figure supplement 2.** Growth profiles (total germ cell volume) of *Drosophila* ovarian follicles.

Gene expression during oogenesis has been previously characterized in purified GSCs (*Kai et al., 2005*; *DeLuca et al., 2020*), downstream follicles (*Sieber et al., 2016*; *Greenblatt et al., 2019*; *DeLuca et al., 2020*), and also in single ovarian cells (*Rust et al., 2020*; *Jevitt et al., 2020*; *Slaidina et al., 2020*; *Slaidina et al., 2021*) or nuclei (*Li et al., 2022*). Early germ cells express conserved germline genes (CGGs) (*Fierro-Constaín et al., 2017*) encoding essential Piwi-piRNA pathway components that repress transposon transcription and promote transposon mRNA slicing and piRNA amplification in nuage (reviews: *Huang et al., 2017*; *Czech et al., 2018*; *Wang and Lin, 2021*). Many important steps in germ cell differentiation are controlled at the translational level (*Slaidina and Lehmann, 2014*; *Ma et al., 2017*; *Kong et al., 2019*; *Blatt et al., 2020*; *Mercer et al., 2021*). For example, the GSC differentiation factor Bam stimulates exit from the stem cell state by repressing high-level ribosome production in the nucleolus (*Zhang et al., 2014*; *Sanchez et al., 2016*; review: *Breznak et al., 2023*) and by lifting the Nanos-controlled translation block of differentiation genes (review: *Mercer et al., 2021*).

GSCs and developing follicles sense environmental nutrients using insulin/IGS and steroid signaling to maximize oocyte output (*LaFever and Drummond-Barbosa, 2005*; review: *Drummond-Barbosa, 2019*). An important regulator of resource adaptation is *Myc*, whose translation becomes derepressed as early as the four-cell cyst (*Harris et al., 2011*) and increases further under favorable conditions starting in region 2b (*Wang et al., 2019*). As in other rapidly growing systems, *Myc* plays an essential role (*Maines et al., 2004*) by controlling hundreds of target genes (*Wang et al., 2019*) that regulate both ribosome production and metabolism.

Here, we extend our knowledge of GSC chromatin modifications and gene expression. Upon leaving the GSC state, daughters form germline cysts and upregulate H3K9me3-rich heterochromatin on transposon sequences in centric chromosome regions. A cyst or syncytium likely protects against transposable element (TE) activity by sharing piRNAs, whereas heterochromatin formation blocks TE transcription. Polycomb complexes dependent on the H3K27 methylase enhancer of zeste (E(z)) repress male germ cell and somatic cell genes that are transiently active in GSCs and early daughters. Metabolism is reprogrammed downstream from GSCs to enhance the production in mitochondria of biosynthetic precursors for follicle growth. Our results argue that GSCs and many other highly potent stem cells modulate the mechanisms they use to control TEs as the initial steps in their differentiation program.

## Results

### Mapping changes in GSC chromatin and gene expression

Chromatin was characterized in purified germ cells or in follicles (*DeLuca et al., 2020*). GSCs and NCs labeled using Germline-GAL4::UASz-tomato were purified from specific ploidy classes using fluorescence-activated cell sorting (FACS) (*Figure 1B*). GSC, NC, and FC ploidy classes (4C, 8C, etc.), and whole follicles were collected and analyzed by Chip-seq for H3K9me3, H3K27ac, H3K27me3, and H3K4me3. ATACseq (*Buenrostro et al., 2015*) was applied to interrogate changes in chromatin accessibility.

Gene expression (*Supplementary file 1a*) was analyzed by RNA-seq using three independent preparations of purified GSCs. To identify gene expression changes in older NCs, we prepared three samples of RNA from wild type or *E(z)* germline knockdown (GLKD) ovaries (*DeLuca et al., 2020*) isolated less than 8 hr after eclosion. At this time ovaries lack follicles older than S6, and are weighted by mass toward stage 4–6 follicles (OVS6). We also analyzed three samples of adult ovaries from well-fed females whose RNA derives mostly from large stage 9–14 follicles undergoing vitellogenesis and eggshell production (WTOV). Finally, we utilized gene expression data from three preparations of mature follicles ('S14') or from S14 oocytes after removing FCs by collagenase digestion and washing ('S14nfc') (*Greenblatt et al., 2019*). Processed data from these experiments are summarized in *Supplementary file 1* as described (Materials and methods; Data availability statement).

Female germ cells do not induce many new genes downstream from the GSC. In fact, the diversity of expressed genes (defined as genes with RNA >1 TPM and SD/mean <0.67) is highest in GSCs and decreases thereafter, while inactive genes increase (*Figure 1C*). A parallel decline is seen in total ATAC peaks, most of which are found around gene promoters and potentially at enhancers (*Figure 1D*). Among transcripts expressed in mature S14 oocytes, 75/79 genetically defined maternal effect genes (95%) and 6520/6597 (99%) total transcripts are already expressed in GSCs, consistent with the importance of translational regulation (*Mercer et al., 2021*). The RNA levels of about 50% of maternal effect genes are already within twofold of their final values at the GSC stage, and 80% increase less than four fold (*Figure 1E*).

### Heterochromatin forms in centric chromosome regions and on transposons downstream from the GSC

Studies of H3K9me3 modification were carried out to look for changes in heterochromatin within early germ cells. GSCs appear to contain less H3K9me3-rich heterochromatin than germ cells within 16-cell cysts based on their lower immunofluorescence staining for the H3K9me3-binding protein HP1a (*Rangan et al., 2011*; *Figure 2A*). Nuclear H3K9me3 staining was visible in GSCs, but levels increase in region 2a and 2b germ cells (*Figure 2B*). Chip-seq experiments for H3K9me3 supported these and earlier observations (*Clough et al., 2007*; *Yoon et al., 2008*; *Rangan et al., 2011*; *Clough*

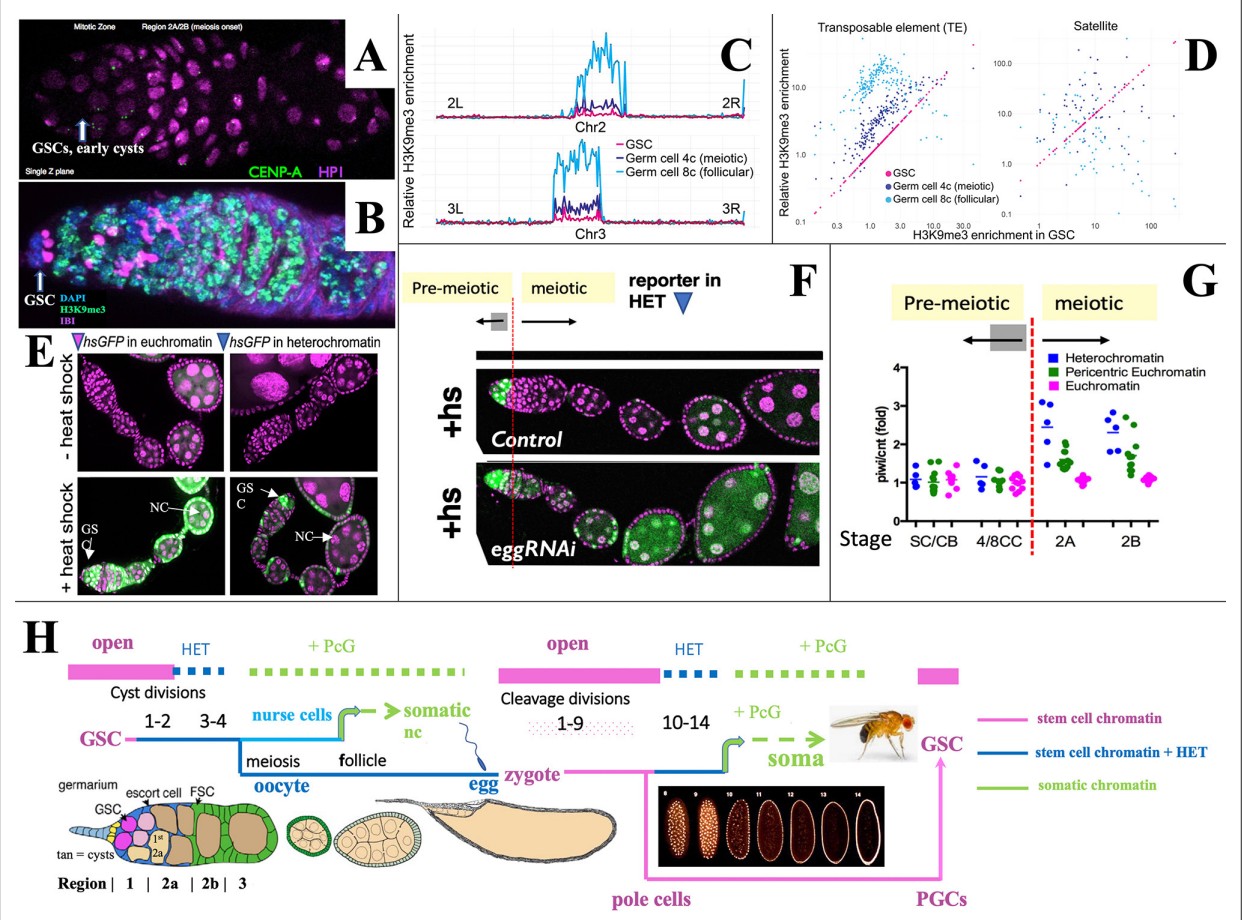

**Figure 2.** Heterochromatin forms in cysts downstream from the germline stem cell (GSC). (**A,B**) Germaria stained for (**A**) H3K9me3-binding protein HP1a (pink) and CENP-A (green), or (**B**) H3K9me3 (green), fusome antibody 1B1 (pink) and DAPI (blue). (**C**) H3K9me3 Chip-seq of unambiguously mapped reads spanning chromosome 2 (lower) and 3 (upper) in 250 kb bins from GSC, 4C nurse cell (NC), or 8C NC. (D) Plots showing relative enrichment of reads mapping to transposable elements (TEs) or to satellite sequences in 4C, and 8C germ cells relative to GSCs. (**E**). A representative euchromatic hsGFP insertion after heat shock expresses GFP (green) in GSCs and nearly all downstream cyst and follicular NC (arrows). In contrast, a typical heterochromatic insertion after heat shock only expresses in GSCs and early cysts (upper arrow), but has become repressed in 16-cell cysts and NC in meiotic and later follicles (lower arrow; other labeled cells are somatic). (**F**) Repression of heterochromatic hsGFP expression after heat shock (+hs) (Control) requires the H3K9 methylase *eggless/SETDB1* (eggRNAi). (**G**) Diagram shows the ratio of hsGFP expression in *piwi*(GLKD) or control (cnt) genetic backgrounds from individual hsGFP insertions (points) at the indicated developmental stages (*Figure 1A*) colored based on insert location in euchromatin (pink), pericentric chromatin (green), or centric heterochromatin (blue). Note: SC = GSC; CB = cystoblast (first cell of germline cyst). (**H**) Summary diagram of *Drosophila* female germ cell generational cycle showing two highly potent stem cell chromatin states (pink) beginning with the GSC or zygote and their downstream developmental trajectories. Both the GSC (maternal) and zygotic trajectories quickly undergo heterochromatin formation (blue) and then somatic cell production dependent on Polycomb repression (green). See text and *Figure 1*, *Figure 1—figure supplement 1* for further details.

*et al., 2014*). H3K9me3 Chip-seq was quantitated in 250 kb bins after screening out ambiguously mapped reads. *Figure 2C* shows that minimal H3K9me3 enrichment relative to chromosome arms was detected in GSCs on chromosome 2 and 3 centric sequences. However, centric H3K9me3 modification increased in 4C NC (NCs and oocytes) and considerably more in 8C NC, which derive from NCs in stage 2–3 follicles. Rising H3K9me3 levels were also found on TE sequences (*Figure 2D*) which are enriched in centric regions, but studies of satellite DNAs were inconclusive. Thus, germ cells begin to increase their centric heterochromatin (HET) as 16-cell cysts form, and H3K9me3 levels rise further during meiotic prophase and early follicular development.

To investigate whether increased H3K9me3 correlates with gene repression, we used the system of individual hsGFP expression reporters that were previously employed to document the onset of Polycomb domain repression in follicular NCs (*DeLuca et al., 2020*). These lines typically do not express

GFP in the absence of heat shock (*Figure 2E*, ' -heat shock') but after a brief standard heat shock, reporters located in euchromatin are readily induced at most germ cell stages (*Figure 2E*, '+heat shock'). In contrast, most insertions located in centric regions were only inducible in GSCs and early cysts, but not in later germ cells (*Figure 2E and F*). This provides evidence that increasing H3K9me3-rich heterochromatin represses gene expression beginning around the time of meiotic onset. When the H3K9 methylase encoded by *eggless (egg)/Setdb1* (*Clough et al., 2007*; *Yoon et al., 2008*; *Rangan et al., 2011*; *Clough et al., 2014*) was knocked down, centromeric reporters could still be expressed in meiotic germ cells and NCs (*Figure 2F*). Reporter gene repression also depended on Piwi. The relative expression of hsGFP in *piwi*[GLKD] compared to control ovaries depended on reporter location and developmental time (*Figure 2G*). Reporters located on chromosome arms (*Figure 2G*, pink) were unaffected by *piwi*[GLKD], whereas only after meiotic onset, those positioned in centric regions (*Figure 2G*, blue) were upregulated in *piwi*[GLKD] compared to control (*Figure 2G*, green), consistent with Chip-seq data (*Figure 2C–D*).

The fact that heterochromatin begins to form downstream from the GSC in germline cysts and repress gene transcription extends a parallel previously noted between stem cell differentiation downstream from the GSC during oogenesis and differentiation downstream from the zygote during early embryogenesis (*DeLuca et al., 2020*). Both take place in a syncytial environment generated either by germline cyst formation or by synchronous embryonic cleavage division, and both include upregulated heterochromatin formation (*Lu et al., 1998*; *Gu and Elgin, 2013*; *Czech et al., 2018*; *Fabry et al., 2021*). *Figure 2H* summarizes the state of germ cell chromatin during the maternal and zygotic portions of the generational cycle. Highly open chromatin characteristic of the GSC state (pink) lasts only briefly before heterochromatin increases (blue). Repression in the oocyte likely becomes widespread as its nucleus condenses to form a karyosome beginning in S3, before meiotic progression resumes about S8. Zygote formation re-establishes stem cell chromatin, which persists during syncytial embryonic cleavage divisions 1–9 and in newly budded pole cells. Only then do cleavage divisions begin to slow, and transposon-directed, Piwi/piRNA-dependent heterochromatin (blue) forms during divisions 10–14 (*Fabry et al., 2021*; *Wei et al., 2021*; *Zenk et al., 2021*). The cellularized embryo goes on to produce diverse somatic cell types using Polycomb repression (green). In a further parallel, follicular ovarian NCs of the GSC lineage repress Polycomb domains and differentiate as somatic cells (*DeLuca et al., 2020*). During embryogenesis the PGCs migrate to the gonads, proliferate during larval stages, and complete the cycle by forming new GSCs at the onset of pupal development (*Bhaskar et al., 2022*; *Grmai et al., 2022*; *Figure 2H*; *Figure 1—figure supplement 1*).

## E(z)-dependent repression contributes to germline- and sex-differential gene expression

We investigated whether some of the more than 900 genes that turn off between the GSC and early follicle stages are repressed by Polycomb. Previous studies identified differences in PRC1 and/or PRC2 (*Figure 3A*) downstream from the GSC (*DeLuca et al., 2020*). GSCs contain high amounts of the alternative PRC2 component Pcl, but relatively low levels of Scm as well as canonical PRC1 and PRC2 subunits. As a result, while H3K27me3 modifications are present on GSC chromatin, site selection by Pcl-dominated PRC2 is stochastic and common foci are largely lacking across all GSCs. After follicles form, Pcl levels fall, Scm levels increase, and more efficient targeting allows canonical Polycomb domains to become repressed in follicular NCs (*DeLuca et al., 2020*). We investigated whether individual genes also undergo PRC2-mediated repression as GSC daughters differentiate.

Genes repressed by E(z) were identified by analyzing GSC-expressed genes that are downregulated in stage 4–6 follicles, but whose RNA persists at higher levels in *E(z)*[GLKD] animals. 423 candidate genes showed either (1) >3-fold increased expression in *E(z)*[GLKD] compared to wild type OVS6 or (2) gene expression after the GSC is downregulated at least 20% less in *E(z)*[GLKD] at S6 compared to OVS6 (*Supplementary file 1b*). Normally, 78% of these genes downregulate more than 10-fold in OVS6 compared to GSC; most no longer express at OVS6 (88%) or in the mature oocyte (98%). Thus, genes regulated in whole or in part through the direct or indirect action of E(z) make up a substantial fraction (40–50%) of the genes that shut off in germ cells between the GSC and early follicular stages.

Some E(z)-downregulated genes function within the GSC itself, and/or in early germ cell development. For example, *bgcn* (*Figure 3B*) facilitates the switch between GSC and cystoblast (*Mercer et al., 2021*). *Bgcn* chromatin shows strong promoter region peaks of ATAC in GSCs and early NCs,

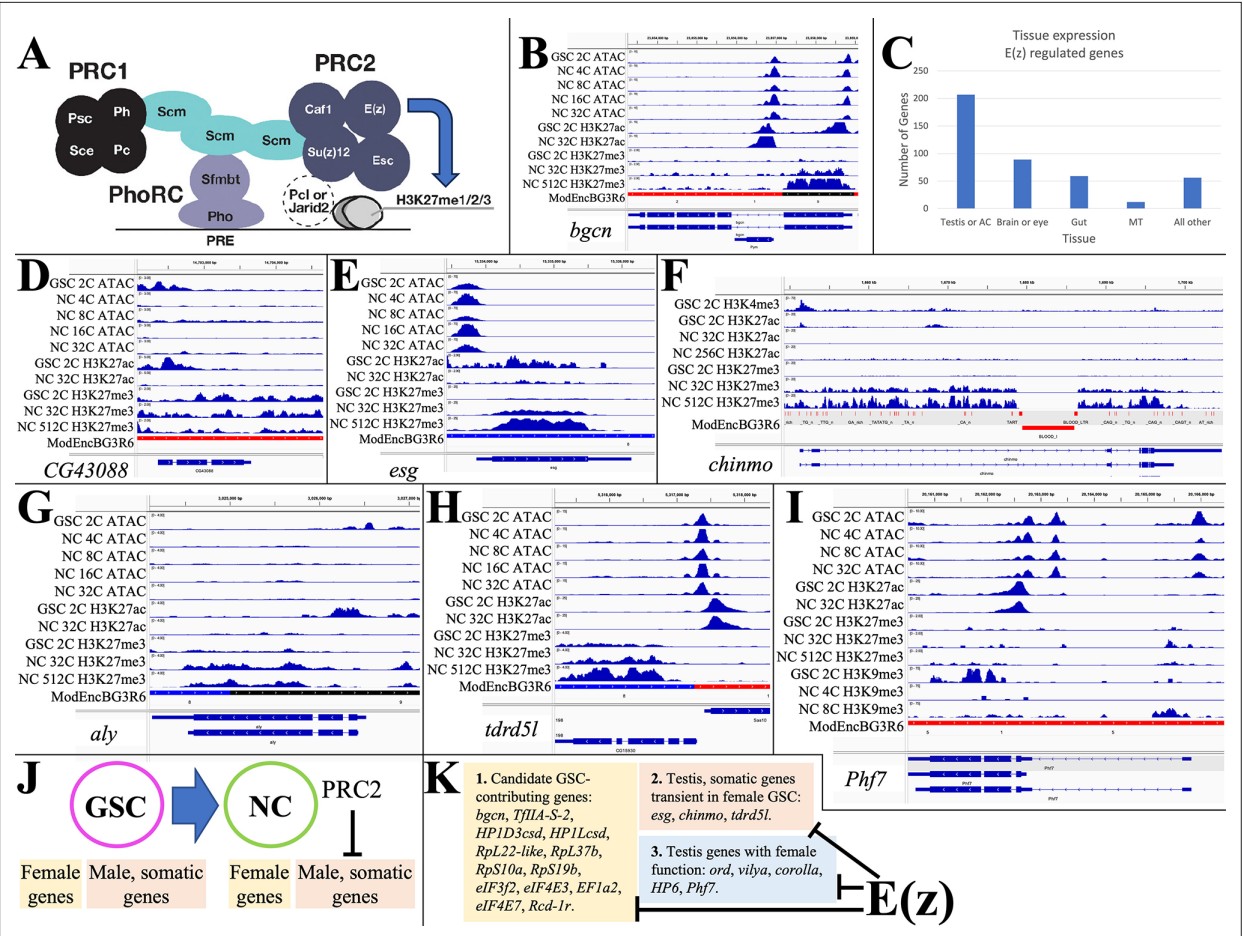

**Figure 3.** E(z)-dependent repression contributes to sex-differential gene expression. (**A**) Diagram of *Drosophila* PRC1 and PRC2. (**B**) Chromatin changes around the germline stem cell (GSC) differentiation gene *bgcn*. The promoter-associated H3K27ac peak is lost by 32cNCs, when H3K27me3 has begun to accumulate. (**C**) The number of genes strongly downregulated by E(z) in young ovaries and their predominant tissue of expression. (**D**) Chromatin changes around the CG43088 testis gene related to a transposon. (**E**) The male GSC gene *escargot (esg)* downregulated by OVS6 shows increased H3K27me3. (**F**) The *chinmo* gene encoding a transcription regulator required in male and female germ cells is repressed as in (**E**). (**G**) The testis regulatory gene *aly* is repressed as in (**E**). (**H**) The tdrdl5 male germ cell regulator is repressed as in (**E**). (I) Phf7 gene regulation. (**J**) Summary of E(z)-dependent repression of male and somatic genes during early female gene expression. (**K**) Three subclasses of genes downregulated by E(z).

and high promoter H3K27ac in GSCs. In 32C and later NCs, H3K27ac has been lost, and H3K27me3 has risen from low GSC levels (GSC: 0.057; 32C NC: 0.584; 512C NC: 0.479; *Supplementary file 1a*), consistent with a direct E(z)-dependent shutoff. Despite a 10-fold increase in *bgcn* RNA levels following *E(z)*[GLKD], S6 expression remains at only 15% of GSC levels, suggesting that *bgcn* is controlled by other pathways as well. Other E(z)-downregulated genes that may function in GSCs encode proteins supporting gene transcription (HP1Dcsh, HP1Lcsd, TfIIA-S-2) or translation (RpL22-like, RpL37b, RpS10a, RpS19b, eIF3f2, eIF4E3, eIF4E7, EF1a2, Rcd-1r) (see *Kong et al., 2019*; *Hopes et al., 2022*).

Analyzing the tissue expression of E(z)-downregulated genes using Fly Atlas (*Leader et al., 2018*) showed that 49% are predominantly expressed in the testis or male accessory gland (*Figure 3C*), including most of the general gene expression isoforms discussed above. Many of the Polycomb-repressed testis-expressed genes such as *CG43088, CG15599,* and *CG43843* present little evidence of ovarian function and some may be recently evolved (*Su et al., 2021*). *CG43088* is related to human harbinger-transpose-derived-1 and shows promoter-associated peaks of ATAC and H3K27ac only in GSCs (*Figure 3D*). The *CG43088* gene body contains significant levels of H3K27me3 in GSCs (0.976) that change little. In *E(z)*[GLKD] early follicles *CG43088* become threefold overexpressed relative to

GSC levels (127±10.2 vs 40±9.46 tpm) and 400-fold overexpressed relative to early wild type follicles (0.28±0.080 tpm).

A much larger group of the testis-expressed genes that are only transiently expressed in early female germ cells are known or likely to have a testis function. *escargot*, encoding a snail class transcription factor needed to maintain male (but not female) GSCs, retains an open promoter in early follicles, but is downregulated (GSC: 91±0.11 tpm; OVS6: 0.070±0.020 tpm), loses H3K27ac, and acquires H3K27me3 at later stages (GSC: 1.54; 32C NC: 16.7; 512C NC: 13.7) (*Figure 3E*). *Chinmo* (*Figure 3F*) encodes a major transcription factor maintaining sexual identity in testis (*Flaherty et al., 2010*; *Grmai et al., 2018*). It undergoes E(z)-dependent shutdown (*DeLuca et al., 2020*) and accumulates H3K27me3 (GSC: 0.753; 32C NC: 11.2; 512C NC: 11.3). *Always early (aly)* is essential for the spermatocyte gene program, has no known female function, and is likewise repressed in early female germ cells supported by E(z) (H3K27me3 GSC: 0.387; 32C NC: 1.25; 512C NC: 0.901) (*Figure 3G*). *Tdrd5l* (*Figure 3H*), a gene implicated in germline male sex regulation (*Primus et al., 2019*), is also strongly expressed in female GSCs (82.4±7.02 tpm), and is downregulated before OVS6 (1.72±0.528 tpm) dependent on E(z) and associated with increasing H3K27me3 (GSC: 0.631; 32C NC: 2.87; 512C NC: 2.30). In addition to controlling many highly testis-enriched genes, E(z) also downregulates other genes with sexually dimorphic expression, including genes preferentially expressed in male accessory gland. However, only about 25% of the male-biased genes affected by E(z) are primarily repressed by Polycomb (i.e. >50%) or show increases in gene body H3K27me3 levels as high as those of Polycomb domain targets, suggesting that the majority of the E(z)-dependent gene repression is indirect.

Some of the 'testis' genes expressed in GSCs function in early female germ cells. *ord, corolla,* and *vilya,* which function in meiosis, are expressed in GSCs and downregulated in young follicles, although the contribution of E(z) to gene downregulation varies: *vilya* (38%), *ord* (20%), and *corolla* (6.6%). *Phf7* (*Figure 3I*) is expressed in GSCs (75.1±9.00), and at higher levels in OVS6 in *E(z)*[GLKD] (62.8±7.15) compared to control (32.4±3.51). However, *Phf7* continues to be expressed throughout oogenesis (S14nfc: 99.3±5.36). Thus, E(z) represses a variety of genes downstream from GSCs in female germ cells (*Figure 3J–K*), where it contributes to normal sex- or tissue-specific gene regulation (*Figure 3J–K*).

## Maintenance of the GSC state

High levels of gene expression are often associated with promoter H3K27 acetylation (H3K27ac) catalyzed by CREB-binding protein (CBP), a conserved histone acetyl transferase (*Holmqvist and Mannervik, 2013*; reviewed in *Lehrmann et al., 2002*). We selectively knocked down the *Drosophila* gene encoding CBP, *nejire (nej)*, in germ cells using specific GAL4 drivers (*DeLuca et al., 2020*). While GLKD of the *white* gene in control ovaries had no effect, knockdown of *nej* in germ cells greatly reduced H3K27ac staining, and arrested follicle growth and development beyond stage 1 (*Figure 4A*).

The chromatin state of several genes highly expressed in the GSC including the GSC differentiation gene *bgcn* (*Figure 3B*) and the core germline gene *piwi* (*Figure 4B*) show strong ATAC peaks in the vicinity of their promoters, high levels of transcription start site (TSS)-associated H3K27ac, and low GSC levels of repressive H3K27me3. To see if this is generally true of GSC genes, we plotted the entire GSC transcriptome as a function of mRNA abundance and H3K27ac (*Figure 4C*). Highly expressed genes usually had high levels of H3K27ac (and low levels of H3K27me3), while genes expressed at low levels did not. H3K27ac peaks in GSCs were almost always (91%) associated with TSSs (*Figure 4D*). Most of these genes also had ATAC peaks near the TSS that accounted for 68% of total ATAC peaks; the remaining ATAC peaks were located in other gene regions, and indicate the location of candidate enhancers. ATAC peaks likely reflect the effects of transcription factor binding at enhancers and promoters, as well as general transcription and splicing factors.

To investigate the GSC transcription program, we identified 101 genes with the highest H3K27ac levels in GSCs (*Supplementary file 1c*). These genes cluster in autosomal pericentric regions, as well as in two euchromatic regions, 31B-F on chromosome 2L, and in region 13F-14C on the X chromosome (*Figure 4—figure supplement 1*). They encode multiple regulators of germ cell development, the cell cycle (including cdk1 and CycD), translation, protein stability, and signaling. Thus, a significant number of genes whose expression is stimulated in GSCs by H3K27 acetylation reside in relatively few clusters, primarily in centric chromosome regions.

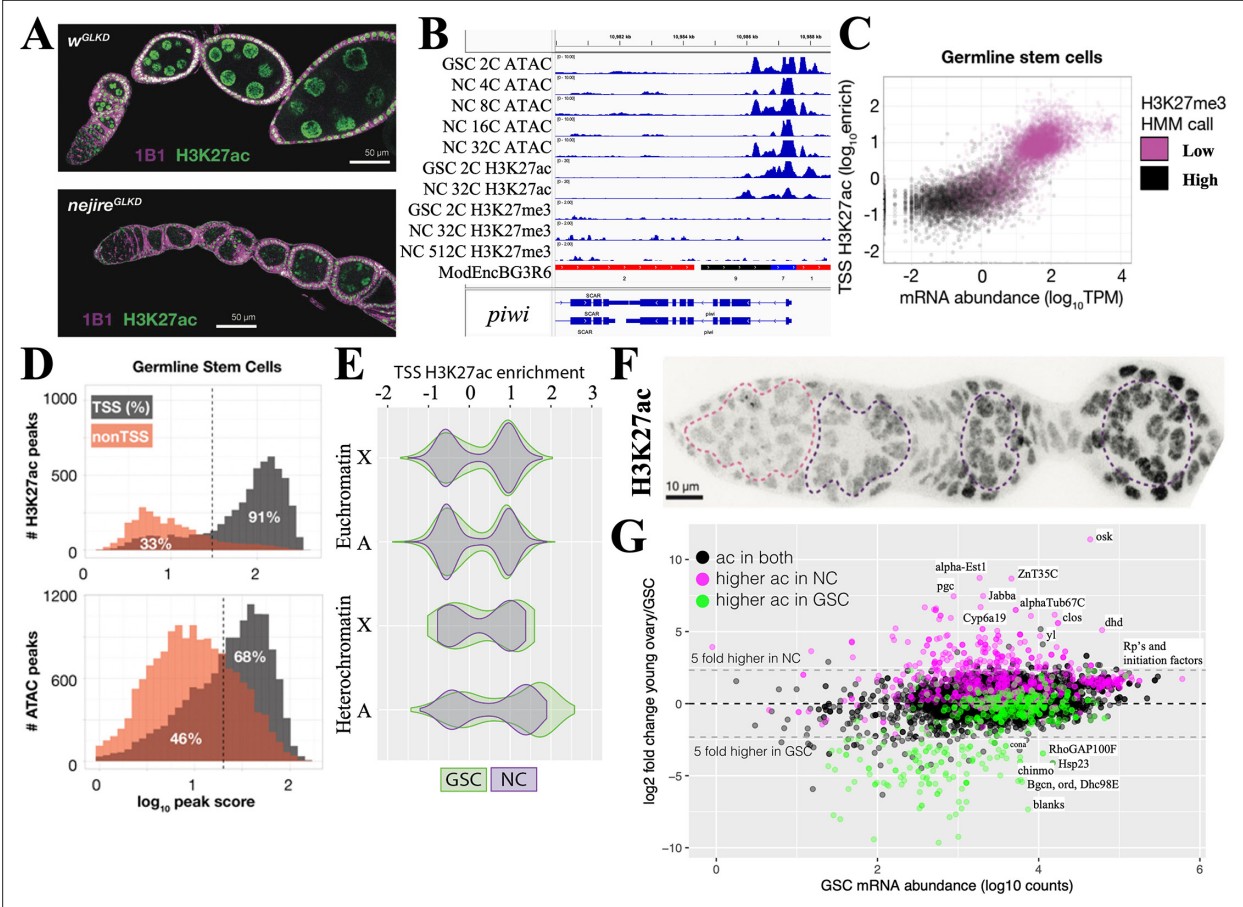

**Figure 4.** Maintenance and exit from the germline stem cell (GSC) state. (**A**) Ovarioles subjected to white gene (*w*GLKD) or *nejire* gene (*nejire*GLKD) germline knockdown. Levels of H3K27ac (green) are greatly reduced in *nej*GLKD and germ cell development is arrested at stages 1–2. Hts staining (1B1, purple) highlights cell membranes and fusomes. (**B**) Chromatin tracks around the conserved germline gene (CGG) *piwi* which is expressed throughout oogenesis. H3K27me3 does not increase. (**C**) mRNA abundance vs the transcription start site (TSS) levels of H3K27ac are plotted versus GSC gene expression. Genes are colored according to hidden Markov model (HMM) calls of H3K27me3 levels. (**D**) The number of gene-associated H3K27ac peaks (upper) or ATAC peaks (lower) in GSCs is plotted versus log$_{10}$ peak score. 81% of H3K27ac peaks and 68% of ATAC peaks are associated with TSSs; the remaining peaks may be associated with enhancer regions. (**E**) Violin plots showing the distribution of H3K27ac enrichment at transcription start sites (TSS) of genes in GSC (green) and nurse cells (gray). Plots are subdivided by gene location in euchromatin (upper) or heterochromatin (lower), and chromosome ( X or autosome (A)). (**F**) Germarium immunostained for H3K27ac showing low levels in early germ cells (dashed pink circle) and increase in S1-S3 nurse cells (purple dashed circles). (**G**) Plot of GSC mRNA abundance vs expression change in young ovary compared to GSCs. A strong correlation can be seen between genes with changes in mRNA levels and changes in H3K27ac levels. Genes (dots) are colored according to changes in ac = H3K27ac levels.

The online version of this article includes the following figure supplement(s) for figure 4:

**Figure supplement 1.** Genes with high H3K27ac levels in germline stem cells (GSCs) cluster in heterochromatin and two euchromatic domains.

## Young NCs metabolically transition

Cells that develop from highly potent stem cells often alter their metabolic state (*Rafalski et al., 2012*; *Hayashi et al., 2017*; *Hayashi and Matsui, 2022*). Downstream from GSCs in young follicles, we identified more than 600 genes that are increased at least threefold (*Supplementary file 1d*). H3K27ac is elevated in NC nuclei from young follicles (*Figure 4F*) and at the promoters of many upregulated genes (*Figure 4G*, pink). Conversely, the expression of genes with reduced acetylation in NCs (green) decreased or changed little. Genes acetylated at similar levels in both cell types (black) rarely changed expression.

Consistent with previous work (*Wang et al., 2019*), Myc is among the transcription factors found to increase sharply in growing follicles (OVS6/GSC: 5.1-fold; *Figure 5A*; *Figure 5—figure supplement 1*). FoxO, the mediator of insulin signaling, is also much higher than in GSCs (7.6-fold). Mitochondrial

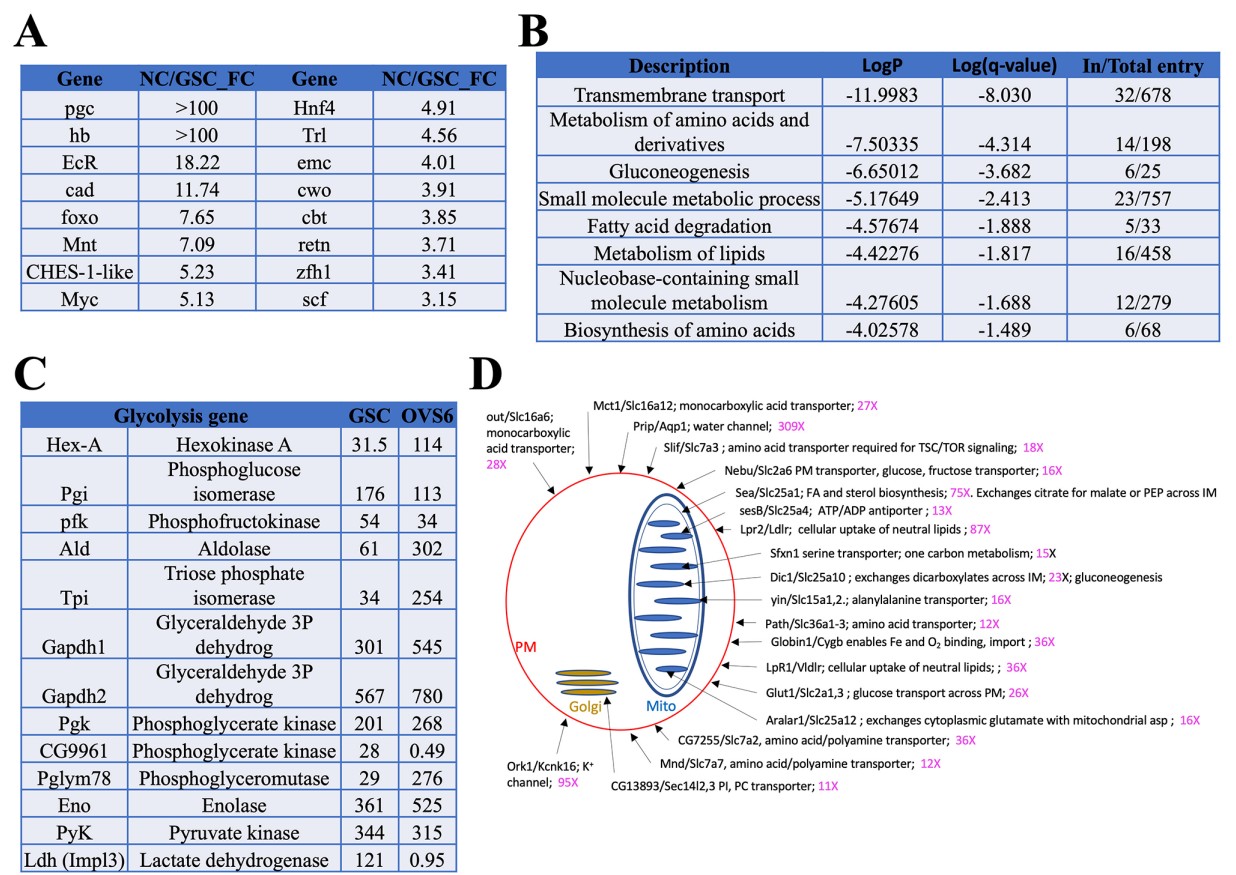

**Figure 5.** Genes induced in germ cells downstream from germline stem cells (GSCs) mediate a metabolic and growth transition. (**A**) Germ cell-expressed transcription regulators with the largest fold change increases in young nurse cell (NC) (OVS6) compared to GSCs. (**B**) Gene ontology (GO) terms for genes upregulated in young ovaries vs GSCs. The GO categories reveal that a dramatic metabolic shift occurs between GSCs and OVS6 NCs. (**C**) Expression values (tpm) in GSCs and OVS6 NCs of glycolytic genes showing mostly upregulation but a shutoff of Ldh. (**D**) Diagram showing upregulated transmembrane transport genes (first GO category in B). Sub-cellular localization of a sample (20) of membrane transporters (Dros name/Mouse name) is indicated (arrows) on the cell drawing showing plasma membrane (red), mitochondrion (blue), or Golgi (orange) along with their fold change (increase) in NC. Many are key regulators of cellular import or mitochondrial metabolism that help generate the precursor nucleotides, lipids, and carbohydrates needed for rapid cellular growth. For complete list see ***Supplementary file 1e***.

The online version of this article includes the following figure supplement(s) for figure 5:

**Figure supplement 1.** Analysis of chromatin changes at enhancers of *Myc* during oogenesis.

oxidative phosphorylation is substantially increased beginning in region 2b (***Wang et al., 2019***). Previous genetic studies also found that ATP synthase function during oogenesis is largely independent of ATP production (***Teixeira et al., 2015***).

We looked for additional metabolic changes by analyzing the genes in ***Supplementary file 1d*** using the gene ontology (GO) program Metascape (***Zhou et al., 2019***; ***Figure 5B***). All the top GO categories suggest that NC mitochondrial metabolism undergoes reprograming in young NCs to upregulate production of precursor molecules including amino acids, nucleotides, carbohydrates, and lipids to support high levels of growth, changes characteristic of aerobic glycolysis. Consistent with upregulation of fatty acid production, we observed increases in acetyl CoA carboxylase mRNA (*ACC*, 3.1-fold) which catalyzes production of malonyl-CoA, the rate-limiting substrate for fatty acid biosynthesis. NC gene expression shows other indications of increased lipid metabolism (see ***Figure 5B***, 'metabolism of lipids'). *CG3902*, encoding an Acyl-CoA dehydrogenase the first step in mitochondrial fatty acid beta-oxidation, is upregulated more than 400-fold. RNA levels increase for both *Lpr1* (12-fold) and *Lpr2* (87-fold). *Hnf4*, encoding a nuclear hormone receptor that senses fatty acid levels, increases 4.9-fold. In mammals, Hnf4 is considered a master regulator of the liver, where it induces

genes (including *FoxO*) involved in lipid metabolism, gluconeogenesis, and amino acid metabolism (*Sieber and Spradling, 2017*), processes that also appear to be upregulated in NCs.

Carbohydrate metabolism is also likely to be strongly increased in young NCs. Sugar transporters are sometimes direct *Myc* targets, and in young NCs *Glut1* sugar transporter expression increases 26-fold, as does *sut1* (5.0-fold). Glycolysis genes often upregulated in *Myc*-expressing cells are also increased (*Figure 5C*). These include genes encoding hexokinase (*Hex-A*, 5.1-fold), triose phosphate isomerase (*Tpi*, 10.5 fold), and phosphoglyceromutase (*Pglym78*, 14-fold). Myc may also increase glutamine uptake and metabolism, which synergizes with increased glycolysis to maintain TCA cycle flux and stimulate lipid metabolism. *Ldh* mRNA production in young NCs is virtually shut down (*Ldh*, 0.011-fold), suggesting that glycolysis is not needed to generate ATP and that lactate is not being generated from pyruvate in large amounts. This would be consistent with upregulation of gluconeogenesis for carbohydrate production (*Figure 5B*). In addition, increased expression of glyceraldehyde-3-phosphate dehydrogenases (*Gapdh1*, 2.5-fold and *Gapdh2*, 1.9-fold) may provide an alternative way to prevent excess NADH generated by glycolysis from accumulating (*Li et al., 2019*). During larval development the metabolic switch to aerobic glycolysis is mediated by the 'estrogen-related receptor' encoded by *ERR* (*Tennessen et al., 2011*), but in young NCs *ERR* levels were unchanged (0.96) relative to GSCs.

A closer examination of 20 of the more than 40 transmembrane transporters that were upregulated at least 10-fold (*Supplementary file 1d*) is shown in *Figure 5D*. Each of these genes has a strong mammalian ortholog. For example, *Sfxn1-3* (upregulated 15-fold), whose mouse ortholog is *Sfxn1*, encodes the mitochondrial serine transporter located in the mitochondrial inner membrane. Serine import by Sfxn1 supplies the sole substrate for one-carbon metabolism, which generates *S*-adenosyl methionine (using *Sam-S*, upregulated 28-fold), glutathione, and NADPH that are needed for methylation, nucleotide biosynthesis, gluconeogenesis, and many other processes (*Kory et al., 2018*). The *Dic1/Slc25a10* transporter (23-fold increased) encodes another inner membrane protein. It catalyzes exchanges of dicarboxylates for inorganic phosphate and sulfur-containing compounds in support of gluconeogenesis. The changes in substrate import alone (*Figure 5D*) reveal how extensively young NCs remodel metabolism downstream from the GSC in support of rapid germ cell growth.

## Discussion

### *Drosophila* female GSCs: a model in vivo totipotent stem cell

Several new insights into highly potent stem cells resulted from these studies using *Drosophila* ovarian GSCs as a model (*Supplementary file 1a*). (1) Polycomb contributes to repressing a class of genes expressed in GSC daughters for no apparent reason. (2) Genes encoding many proteins needed and expressed by stem cells are clustered in centric heterochromatic domains and highly modified with H3K27ac in GSCs. (3) A novel mechanism of transposon control used by stem cells is implicated by the initial focus of their daughter cells on syncytial interconnection and H3K9me3-mediated repression. (4) Animal life cycles require two connected totipotent differentiation processes, rather than just one. The ability to study totipotent GSCs and genetically manipulate their operating instructions will further advance our understanding of these topics. The resources presented here (*Supplementary file 1*) expand genomic data on GSCs and extend developmental studies of their daughters that are highly annotated (Flybase: *Gramates et al., 2022*).

### E(z)-dependent repression downstream from the female GSC modulates sex- and tissue-specific gene expression

GSCs transiently express more than 400 diverse genes whose downregulation depends on E(z), including ~50% that are primarily found in male germ cells (*Supplementary file 1b*). The functional significance of their Polycomb-mediated repression, and the pathways by which many targets are indirectly controlled, remain to be determined. Polycomb repression is essential for follicular NC differentiation, for follicle development beyond about S5, and for female fertility (*DeLuca et al., 2020*). Early repression of the genes in *Supplementary file 1b* may also be important without arresting oocyte development before S5.

The Polycomb-associated downregulation of genes in early *Drosophila* female germ cells may be analogous to Polycomb regulation during mammalian PGC development (review: *Loh and Veenstra,*

*2022*). Following induction in the proximal epiblast at about E7.0, mouse PGCs migrate to the gonad which they reach between E10.5 and E11.5. During migration, the PGCs significantly reduce their DNA methylation and repressive H3K9me2, but upregulate H3K27me3 (*Seki et al., 2005*). The timing of sex-specific PGC differentiation depends on the PRC2 component EED, which may interact with Dnmt1 to regulate this process (*Lowe et al., 2022*). During part of this period they arrest in G2 and modify RNA polymerase tail phosphorylation to repress transcription (*Seki et al., 2007*), changes that also take place in migrating *Drosophila* PGCs (*Su et al., 1998*; *Deshpande et al., 2004*; *Martinho et al., 2004*). These events allow mouse PGCs to reacquire totipotency, reactivate conserved germ cell genes such as *Ddx4*, and to silence gene expression resulting from somatic signals received prior to their reprogramming as germ cells (*Hill et al., 2018*). Polycomb-dependent silencing in *Drosophila* may likewise 'clean up' extraneous transcription to compensate for stochastic or incomplete programming at earlier stages.

Our finding that GSCs and early female germ cells normally express many male germ cell genes calls into question elevated male gene expression as a criterion for identifying germline sex determination genes (*Bhaskar et al., 2022*; *Grmai et al., 2022*). Mutants that increase testis-related gene expression and also arrest female germ cell development or generate GSC-like tumors may not determine germline sex, because elevated male gene expression is normal in female germ cells at early stages. For example, overexpression of the proposed germline sex determination regulator Phf7 (*Yang et al., 2012*) in female germ cells disrupts their differentiation and induces expression of several hundred testis genes (*Shapiro-Kulnane et al., 2015*; *Smolko et al., 2020*). We find that these genes extensively overlap the testis genes normally downregulated by E(z). Mutations of *stonewall (stwl)*, encoding a heterochromatin-interacting factor required to maintain GSCs (*Clark and McKearin, 1996*), also disrupt differentiation, alter germline gene expression, and overproduce Phf7 (*Zinshteyn and Barbash, 2022*). The testis genes upregulated by *stwl* mutation strongly overlap those found in normal female GSCs, which itself 'overproduces' *Phf7* transcripts.

The SKI complex is required for follicle development beyond early follicle stages and degrades some early female germ cell RNAs (*Blatt et al., 2021*). We find that about 10% of SKI-complex target genes overlap with the E(z)-repressed genes, including *RpL22-like* and *dany*. However, most SKI-complex target genes are not even expressed in GSCs (tpm <1) and the reason for this small overlap is unknown.

## A connection between totipotent stem cells and genes in heterochromatic domains

GSCs express many important genes residing in centric domains that display high levels of promoter-associated H3K27 acetylation, suggesting they may share some regulation in common (*Supplementary file 1c*). *ovaries absent (ova)* is needed for heterochromatin formation and transposon repression in GSC daughters (*Yang et al., 2019*). *vasa* controls mRNA translation in female germ cells and represses TEs in nuage. *me31B* is required to form germ cell P granules (*Nakamura et al., 2001*), while insulin pathway genes *chico, Pten,* and *Reptor-BP* couple GSC activity to nutrients. *daughterless* is maternally required for sex determination (*Cronmiller and Cline, 1987*). One of the two 'euchromatic' zones housing genes in this group, located in region 31, is connected to heterochromatin. This region forms the unusual 'gooseneck' region of polytene chromosomes and binds high level of HP1a (*James et al., 1989*; *DeWit et al., 2007*). Many conserved genes in this domain are located in heterochromatin in *Drosophila virilis* and *Drosophila pseudoobscura* (*Caizzi et al., 2016*). This chromatin context in cells with relatively low H3K9me3 levels may be advantageous for genes with critical functions in stem cells, and may facilitate access by CBP and transcription factors. Some germline-specific genes located in H3K9me-rich chromatin in *C. elegans* are also distinctively regulated (*Methot et al., 2021*; *Padeken et al., 2022*).

## Formation of germline cysts and heterochromatin downstream from totipotent stem cells suppresses transposon activity

The formation of syncytial germline cysts downstream from GSCs or PGCs in the male and female gonads of diverse species helps protect gametes from meiotic drive and allows cyst cytoskeletal polarity to specify which female cyst cells will be oocytes and which will be nurse cells (review: *Spradling et al., 2022*). However, the syncytial environment in the germline cyst, whose cells express

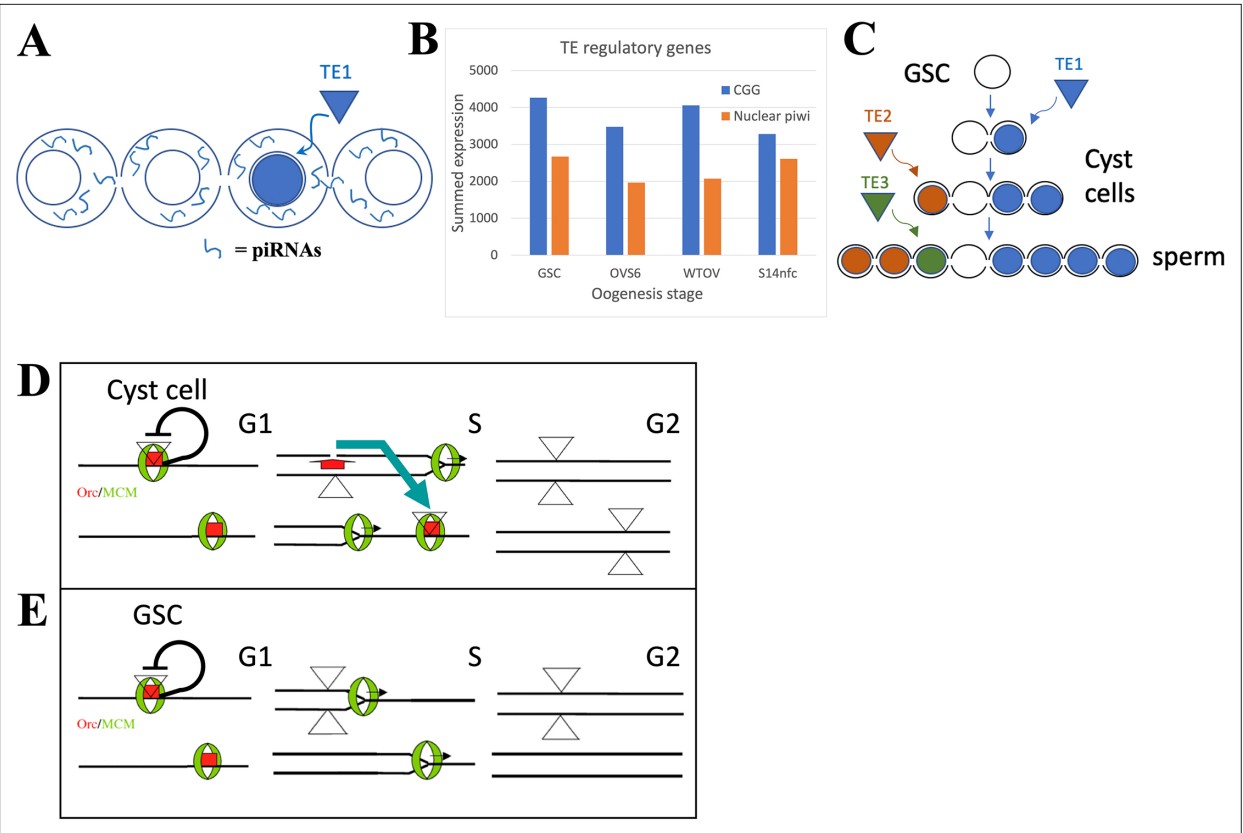

**Figure 6.** The germline stem cell (GSC) state provides high immunity to transposable elements (TEs). (**A**) Proposed amplification of TE resistance by a germline cyst or syncytial stage downstream from a highly potent stem cell by amplifying and sharing piRNA (blue lines) between all connected cells following TE1 movement in one cell. (**B**) Summed expression (tpm) of conserved germline genes (CGGs) (blue) involved in transposon regulation and nuclear piwi genes (orange). (**C**) Illustrative diagram of cluster analysis in a male GSC lineage with two cyst divisions and a single meiotic division leading to eight sperm. The number of sperm with identical TE insertions depends of the time of insertion during cyst formation, as shown by three different TE examples that insert at different stages. (**D**) Model of P element (triangles) copy number increase on chromosomes (horizontal lines) during a cyst cell or meiotic germ cell cycle by 'replication timing'. Red = ORC proteins, green = MCM proteins, black line shows postulated repression prior to loss of pre-replication complex in S phase by origin activation or fork passage. Blue arrow shows transposition from a replicated to an unreplicated region recognized by unfired pre-replication complex during later S phase (modified from **Spradling et al., 2011**). (**E**) Like (**D**), but in a GSC whose short G1 is proposed activate all origins simultaneously.

high levels of piwi/piRNA pathway components (**Figure 6B**), can also help control TE activity. The presence of a cyst or syncytium would allow piRNAs induced by TE activity in one cyst cell to be shared with the other cyst cells (**Figure 6A**), enhancing resistance to that TE in the other cells before transposition can occur there (reviews: **Huang et al., 2017**; **Czech et al., 2018**; **Wang and Lin, 2021**). The high similarity of sister cells recently derived from a common precursor likely enhances the efficacy of such sharing.

Activation of the nuclear arm of the piRNA pathway contributes to the H3K9me3-modified chromatin that increases in GSC daughters as they prepare and enter meiosis. These increases utilize *piwi* and *eggless* and target transposon-related as well as uniquely mapping sequences (**Figure 2F and G**; **Clough et al., 2007**; **Clough et al., 2014**; **Ninova et al., 2020**; **Chen and Aravin, 2023**). Very similar heterochromatin formation takes place during the last few syncytial cleavages of the *Drosophila* embryo downstream from the zygote (**Lu et al., 1998**; **Gu and Elgin, 2013**). Heterochromatin formation during early embryogenesis in *Drosophila miranda* also involves piRNA-directed H3K9me3 deposition that may start even earlier in the cleaving pre-blastoderm embryo (**Wei et al., 2021**). These embryos remain syncytial and likely share piRNAs between preblastoderm nuclei, prior to cellularization. Similar pathways of heterochromatin formation act in diverse organisms (**Liu et al., 2014**; **Onishi et al., 2021**).

The formation of interconnected germline cysts or syncytia just downstream from highly potent stem cells has been extensively conserved in male and female germ cells throughout animal phylogeny (review: *Spradling, 2023*). These events can be traced back to the highly potent stem cells (i.e. interstitial stem cells [ISCs], neoblasts) found in basal animals including hydra, planaria, parasitic flatworms, and other organisms (*Zeng et al., 2018*; *Issigonis and Newmark, 2019*; *Reddien, 2022*; *Littlefield, 1994*). Stem cells in basal eukaryotes express largely the same set of CGGs present in bilaterian animal germ cells (*Fierro-Constaín et al., 2017*), and frequently also form cysts downstream from highly potent germline and somatic stem cells. Notably, cysts similar to those produced by *Drosophila* GSC daughters not only arise downstream from hydra GSCs (*Littlefield, 1994*), but immediately downstream from somatic ISC daughters which generate the cnidocytes required for feeding (*David, 2012*). The intercellular bridges (IBs) in cnidoblast cysts resemble germline IBs and were observed to share cytoplasm between cyst members (*Fawcett et al., 1959*). Hydra in ISCs, GSCs, and some downstream cells express *piwi* genes carrying piRNAs that mostly target transposons (*Lim et al., 2014*), suggesting that cnidoblast cysts also help control TE activity within and downstream from highly potent somatic and germline stem cells.

## GSCs and highly potent stem cells contain an enhanced system of transposon resistance

The timing of cyst and heterochromatin formation just after the stem cell state is surprising. Potent stem cells and GSCs are collectively immortal and responsible for protecting a species' genome from TEs over developmental time, spanning potentially millions of generations. Animal genomes contain diverse TEs, that in many groups now constitute a substantial fraction of the total (review: *Wells and Feschotte, 2020*). Stem cells and germ cells would seem to require the highest levels of TE defense consistent with their high expression of CGGs. Yet female *Drosophila* GSCs display relatively open chromatin (*DeLuca et al., 2020*) and like the *Drosophila* zygote only form a syncytial environment and upregulate heterochromatin in downstream cells. The timing of these events argues that stem cells use additional mechanisms in addition to the cytoplasmic *piwi* pathway, for transposon control, that render H3K9me3-mediated heterochromatin formation unnecessary. However, this stem cell repression mechanism must fade quickly in daughter cells, necessitating them to increase alternative transposon defense mechanisms.

Does evidence exist that transposition is relatively low in totipotent stem cells, but increases in early derivative cells? Cluster analysis, in theory, can measure when a TE moves during germ cell development. In the simplified lineage (with four-cell cysts and one meiotic division) shown in *Figure 6C*, the TE1 (blue) insertion at the two-cell cyst stage will generate four sperm with the same genetic change, whereas the TE2 (orange) insertion at the four-cell stage will induce two identical mutant sperm, and the TE3 (insertion) at the sperm stage will result in one unique insertion. In practice, the low rate of transposon movement in animals during normal development makes such studies difficult.

A *Drosophila* genetic screen (*Karpen and Spradling, 1992*) generated single P element transposition events in the progeny of individual males that were subsequently analyzed to determine cluster size. 100–200 progeny were analyzed from each of ~15,000 individual males. About 6800 had zero transposon-containing progeny, 6173 had one offspring with a jump, 1024 had two such progeny, 17 had three, and 2 had four. Progeny with transpositions were then analyzed by enhancer trapping and limited insertion sequencing, indicating that overall about 7909 transpositions occurred singly and that 367 occurred as doublets. From the 19 males with three or four transposed offspring, only two contained a single sibling with one identical transposition; thus no clusters larger than two were observed. If P elements had jumped during preadult germ cell development or at the stem cell stage, then each mutant stem cell (from 10 to 20 total) would have generated clusters amounting to 5–10% of progeny, that is 5–20 offspring with identical insertions. The fact that only an estimated 25–50% of sperm (randomly) survive in recipient females to fertilize an egg further increases the ability to detect GSC insertions. These results argue that P elements transpose much less frequently at or before the stem cell stage, but show that transposition has begun before the end of cyst formation.

Consistent with a loss of effective stem cell repression, transposon activity increases downstream from the zygote. TE transcription is high in early *Drosophila* embryos (*Scherer et al., 1982*) and elevated TE expression and activity are widely observed in early mouse and human embryos (*Xiang and Liang, 2021*; reviewed in *Friedli and Trono, 2015*). Particular classes of retroelements drive

expression in temporal and tissue-specific patterns (*Göke et al., 2015*). Programmed changes in TE expression patterns can accurately indicate the developmental state of stem cell lines that capture stages downstream from full pluripotency (*Theunissen et al., 2016*). TE-dependent transcription in early embryonic cells may even be functionally important (*Percharde et al., 2018*). An association of intrinsic TE resistance with high potency states might also explain why DNA methylation levels can be reduced, as during the reprogramming of PGCs back to a pluripotent state to generate gonocytes (*Hill et al., 2018*).

## Altered cell cycle and DNA replication behavior may enhance stem cell transposon resistance

The most likely sources of intrinsic TE resistance in potent stem cells are their unusual cell cycles. Such stem cells and their immediate progeny in gonads, embryos, and cell lines frequently have very short G1 and long G2 phases (*Hsu et al., 2008*; *Sheng and Matunis, 2011*; *Pauklin and Vallier, 2013*; *Ter Huurne and Stunnenberg, 2021*; *Ables and Drummond-Barbosa, 2013*; *Seller et al., 2019*; *Vogg et al., 2021*). DNA replication during S phase is also altered, due to changes in the number and order of origin activation (*Hyrien et al., 1995*; *Ferreira and Carmo-Fonseca, 1997*; *Seller and O'Farrell, 2018*; *Macheret and Halazonetis, 2018*; *Rausch et al., 2020*; *Klein et al., 2021*), replication elongation rate (*Nakatani et al., 2022*), or because of other changes in DNA replication and repair (*Wooten et al., 2019*; *Sjakste and Riekstiņa, 2021*). These differences may affect TE movement because many transposons are sensitive to cellular replication. P elements, for example, insert preferentially near replication origins (*Spradling et al., 2011*; *Cao et al., 2023*) and this may allow them to transpose after duplication to a new unfired origin, often in a late replicating region (*Figure 6D*). Altering replication timing to eliminate late S phase might disrupt this pathway by eliminating available targets before they can be used (*Figure 6E*). Thus, highly potent stem cells may alter cell cycle and replication properties to enhance their ability to suppress TE movement.

Differences between highly potent stem cells and later more differentiated derivatives frequently involve the loss of late replication. Early in *Drosophila* cleavage there is no late S phase, but after cycle 9, derepression of Rap1-binding factor (Rip1) allows Rip1 to bind to and repress satellite DNA duplication, prolonging S phase (*Seller and O'Farrell, 2018*). Pericentromeric heterochromatin replicates in early to mid S phase in mouse ES cells, but replicates late upon differentiation, a change that is associated with reduced acetylation of H3K9 and H3K8 in these regions (*Rausch et al., 2020*). The reason DNA replication is temporally regulated is not understood although it is proposed to facilitate chromatin epigenetic regulation (see *Klein et al., 2021*). Why differentiated cells acquire a distinct late S phase also remains a mystery, despite its apparent universality. Finally, it is not at all clear that avoiding late replication would be sufficient to reduce TE movement enough to maintain genomes over evolutionary timescales. However, the striking organization of eukaryotic genomes in two compartments, one of which replicates late in S phase and houses most genomic transposons, is consistent with a strong TE requirement for late S phase.

Several other properties of highly potent stem cells may contribute to TE resistance. Despite a relatively open chromatin state, highly potent cells like *Drosophila* female GSCs might exclude expression of transcription factors that recognize and activate TE promoters. The special isoforms of general transcription, splicing, and translation machinery expressed by GSCs (*Kai et al., 2005*; *Shigenobu et al., 2006*) might interfere with TE gene expression. Except for RpS5B (*Kong et al., 2019*), production of most of these isoforms shuts off downstream from female GSCs (*Figure 3K*, *Supplementary file 1b*).

## Animal generational cycles involve sequential germline and zygotic programs of stem cell differentiation

The process of transitioning from a stem cell to a more differentiated state represents a fundamental unit of development. Our studies emphasize that two such stem cell differentiation units are required for each generational cycle, rather than one as often visualized (*Figure 2H*). Maternal and zygotic stem cell development act over parts of two animal generations (zygote to zygote) to complete a single generational cycle (GSC to GSC). While each unit involves common regulatory events, including exit from a totipotent stem cell state, chromatin remodeling, growth, and reprogramming of metabolism, there are many important differences between the maternal and zygotic cycles that make both essential. The embryo cannot overcome or replace all the events that took place in its mothers' germline.

Restoring potential immortality likely represents the major focus of the maternal stem cell program. Rejuvenation occupies most of oogenesis, but continues during zygotic development in the embryo. Perturbations in metabolism during oogenesis can have lasting effects on the zygote (*Hocaoglu et al., 2021*). Maternal Piwi is essential for normal embryonic germ cell development and female fertility (*Gonzalez et al., 2021*). Polycomb function during oogenesis is needed for normal mouse embryonic development (*Posfai et al., 2012*). A better understanding of the processes that link maternal and zygotic pluripotent stem cell differentiation promises to illuminate the full range of mechanisms that sustain animal species and to further improve the cell therapies that depend on such an understanding.

# Materials and methods
## *Drosophila* strains

| Mutant alleles | Source | Stock number | Citation | Other info |
|---|---|---|---|---|
| bamΔ86 | Bloomington Drosophila Stock Center: BDSC NIH P40OD018537 | 5427 | Flybase (*Gramates et al., 2022*) | |
| yl$^{13}$ (K621) | BDSC | 4320 | Flybase (*Gramates et al., 2022*) | yl$^{13}$ v$^{24}$/FM3 |
| yl$^{15}$ (11-380) | BDSC | 4621 | Flybase (*Gramates et al., 2022*) | y$^1$ cv$^1$ v$^1$ yl$^{15}$ f$^1$/FM0 |
| Cap-H2$^{TH1}$ | BDSC | 2608 | Flybase (*Gramates et al., 2022*) | Df(3L)W10, ru$^1$ h$^1$ Cap-H2$^{TH1}$ Sb$^{sbd-2}$/TM6B, Tb$^1$ |
| Cap-H2$^{TH2}$ | BDSC | 12126 | Flybase (*Gramates et al., 2022*) | y$^1$ w$^*$; P{w[+mC]=lacW}l(3)j9A5$^{j9A5}$ Cap-H2$^{TH2}$/TM6B, Tb$^1$ |

| RNAi Lines | Source | Stock number | Insertion site | Vector | Other info |
|---|---|---|---|---|---|
| UASp-E(z)$^{RNAi}$ | BDSC | 36068 | attp40 | Valium22 | TRiP.GL00486 |
| UASp-nej$^{RNAi}$ | ??? | | | | |

| Gal4 Drivers | Source | Stock number | Insertion site | Citation | Genotype |
|---|---|---|---|---|---|
| Nos-Gal4 | BDSC | 4937 | 3R:10407270 | Flybase (*Gramates et al., 2022*) | P{GAL4::VP16-nos.UTR} |
| MTD-Gal4 | BDSC | 31777 | X, 2, 3R:10,407,27 0 | Flybase (*Gramates et al., 2022*) | P{otuGAL4::VP16}; P{GAL4-nos.NGT}; P{GAL4::VP16nos.UTR} |

| Fluorescent Lines | Source | Stock number | Insertion site | Citation | Genotype |
|---|---|---|---|---|---|
| UASz-tdTomato | DeLuca | | VK33 | *DeLuca et al., 2020* | |

## Primary antibody table

| Epitope | Species | Source/reference | Concentration for IF |
|---|---|---|---|
| Hts (1B1) | Mouse | Developmental Studies (*Zaccai and Lipshitz, 1996*) | 1:50 |
| H3K27ac | Rabbit | Active Motif (#39135) Lot 17513002 | 1:1000 |
| vasa | Rabbit | Lab-generated (Final bleed FB87) | 1:1000 |

## Immunofluorescent staining and microscopy

Flies are fed on wet yeast for 3 days before dissection. Dissected ovaries are fixed in 4% PFA and 0.01% PBST for 30 min, blocked with 5% Normal Goat Serum before overnight incubation in primary

antibodies. Please see Primary antibody table for antibody sources and their appropriate dilution. Secondary antibodies are incubated overnight at 1:1000 dilution. DNA is stained with DAPI (0.5 μg/ml) and the ovaries are mounted in 50% glycerol for imaging. Images are acquired using Leica TCS SP5 with 63× oil objective.

## Volume measurement

MTD-Gal4 >UASz-tdTomato flies with labeled germline cells are fed on wet yeast for 3 days. Ovaries are dissected carefully with forceps to break apart the surrounding muscle sheets. Dissected ovaries are fixed in 4% PFA for 10 min, stained with DAPI, and mounted in VECTASHIELD. For yl mutant and control lines, dissected ovaries are stained with anti-vasa to mark germline cell volumes (see immunofluorescent staining method for details). Full z-stacked images of the ovary are taken under 40× oil objective with Leica TCS SP5 Confocal Microscope. Reconstruction of 3D images is performed with Oxford Instruments Imaris software. Volumes and total DAPI signal pixels of individual follicles are recorded and developmental stages of the follicles are manually noted. Growth rate changes are plotted and modeled with R.

## Growth modeling

Local polynomial regression was used to accommodate for the complexity of the data. Smoothing factor = 0.75 and degree of polynomial = 2 were used. The data set is fairly dense with n>20 for each stage and is therefore suitable for local polynomial regression. However, the result of LOESS can be difficult to interpret because it cannot be written down as simple equations (*Guthrie et al., 2002*). To further test the developmental growth rate changes, multiple linear regression was applied with two conditional terms. Condition1 distinguishes follicles before (stages 1–5) and after (stages 6–10) NC chromatin dispersion. Condition2 distinguishes follicles before (stages 1–7) and after (stages 8–10) onset of vitellogenesis. The model to be tested is as follows:

$$\log10 \text{ Volume} = a0^*\log2\text{DNA} + a1^*\log2\text{DNA}^*\text{condition1} + a2^*\log2\text{DNA}^*\text{condition2}$$
$$+ b0 + b1^*\text{condition 1} + b2^*\text{condition2}$$

The fitting result is as follows:

```
## Coefficients:
## Estimate Std. Error t value Pr(>|t|)
## (Intercept) (b₀) 3.47678 0.03855 90.186<2e-16 ***
## log2norm.DAPI (a₀) 0.23343 0.01152 20.263<2e-16 ***
## condition1 (b₁) –0.13196 0.08155–1.618 0.106569
## condition2 (b₂) 1.45582 0.33120 4.396 1.49e-05 ***
## log2norm.DAPI:condition1 (a₁) 0.05044 0.01544 3.266 0.001204 **
## log2norm.DAPI:condition2 (a₂) –0.13115 0.03544–3.701 0.000252 ***
## ---
## Signif. codes: 0 '***' 0.001 '**' 0.01 '*' 0.05 '.' 0.1 ' ' 1
```

## FACS

The protocol is adapted from *DeLuca et al., 2020*. Briefly, females fed on wet yeast are dissected and collagenase treated and fixed in 2% PFA. Fixed nuclei are isolated, stained with DAPI, and filtered before sorted with an a Becton-Dickenson FACSAria III cell sorter.

## ChIP-seq

ChIp-seq was carried out as described in *DeLuca et al., 2020*. Briefly, we standardized input to the number of genomes rather than the number of cells, as we worked with cells with various ploidy levels spanning from 2C to 512C. For each IP, the equivalent of 500,000 2C sorted nuclei were used. Samples were mixed with a specific amount of fixed mouse 3T3 cells as a spike-in if applicable. Mixed samples were then sonicated to acquire mono-nucleosome fragments. Fragment sizes were confirmed with the Bioanalyzer (Agilent). 1% of each sample was set aside and later was reverse cross-linked to be used as input control. The rest of the nucleosome fragments are incubated with antibody-conjugated beads against the corresponding target protein for chromatin immunoprecipitation. Afterward, all samples were treated with RNAse A, and subsequently proteinase K. DNA was

then extracted with phenol/chloroform, precipitated with NaAc/ethanol, washed with 70% ethanol wash, and resuspended samples in 10 µl water and of which all 10 µl was used for library preps. The library is prepared with Takara Bio ThurPLEX DNA seq kit and sequenced for 75 b.p. single-end reads on an Illumina NextSeq 500.

## ATAC-seq

ATAC-seq libraries was originally prepared with the illumina Nextera DNA Sample Preparation Kits. Later experiments used the Illumina DNA Prep kit with a modified protocol. Sorted nuclei are resuspended in the Tagmentation Buffer 1 (TB1) and incubated with DNA Tagment DNA Enzyme (TDE1) (available as illumina Tagment DNA TDE1 Enzyme and Buffer Kits) for 30 min at 37°C. The reaction is then stopped with Tagment Stop Buffer (TSB) at 65°C overnight and digested with RNAase and Proteinase K for 1 and 2 hr, respectively. Tagged DNA is extracted with phenol/chloroform and amplified with Enhanced PCR Mix (EPM), i7 and i5 index adapters for 10 PCR cycles. Amplified DNA is cleaned up with the Sample Purification Beads or alternatively with the Beckman Coulter AMPure XP beads. The finished library is checked for size and concentration with Agilent Bioanalyzer and sequenced for 75 b.p. single-end reads on an Illumina NextSeq 500.

## RNA-seq analysis

75 b.p. single-ended RNA-seq reads are aligned to the *Drosophila melanogaster* release six reference genome (dm6) and Ensembl annotation release 98 (dm6.22.Ensembl.28) with RSEM quantification package (1.3.0) (*Li and Dewey, 2011*) paired with STAR RNA-seq aligner (2.7.3a) (*Dobin et al., 2013*). Differential expression analysis is performed with DESeq2 (1.26.0) (*Love et al., 2014*). Aligned reads are imported into DESeq2 with the tximport package (1.14.2) (*Soneson et al., 2015*). Normalized bigwig data for visualization is generated with deepTools (3.1.3) (*Ramírez et al., 2016*). GO analysis was performed using DAVID (*Huang et al., 2009a*; *Huang et al., 2009b*).

## ChIP-seq analysis

ChIP-seq analysis was performed as described in *DeLuca et al., 2020*. 75 b.p. single-ended ChIP-seq reads were aligned to either the *D. melanogaster* release six genome (w/o spike in) or a hybrid genome of *D. melanogaster* and *Mus musculus* (w/ spike in) using Bowtie2 (*Langmead and Salzberg, 2012*) and processed using SAMtools (*Li et al., 2009*). To remove ambiguously mapped reads, only mapped reads that pass the mapping quality filtering were used for subsequent analysis. The corresponding BedGraph and bigWig files were generated using the bedtools genomecov function (*Quinlan and Hall, 2010*) and kentutils wigToBigWig function (citation), respectively. Differential ChIP-seq analysis was performed with MACS2 bdgdiff function (*Wang et al., 2019*) Unresolved. Signal distribution was modeled with mixtools using CPM normalized read counts (*Benaglia et al., 2009*). The read depth across a genomic region of interest was visualized and presented in the Integrative Genomics Viewer (*Robinson et al., 2011*).

## Chromatin state calling

Bowtie2 was used to map 75 b.p. ChIP and Input reads (--very-sensitive -N 1) from GSCs, NCs, and FCs to the *Drosophila* genome (dm6), and subsequently filtered and sorted using SAMtools 'view' (-q 10 -hF 4) and 'sort' subcommands. BEDTools (*Quinlan and Hall, 2010*). 'makewindows' was used to bin the genome into 500 bp non-overlapping bins (-w 500 -s 500) and the 'coverage' subcommand was used to count the number of reads in each bin (-counts). For each ChIP/Input pair of bedGraphs, bins with 0 reads in either sample were discarded from both. The fold-change for each bin between the ChIP and Input was then taken. Raw fold-change values were between-sample normalized to the median fold-change value, similar to DEseq2 (ref), and then $\log_{10}$ transformed. A three-state hidden Markov model was trained on the $\log_{10}$ FC ratios using PufferFish v0.1.20200925 (https://github.com/JohnUrban/pufferfish; commit. f0c862a122357c062316d833f58ed6f6955c141c). It was initialized by learning the Viterbi Encoded state path starting with uniform initial and transition probabilities and semi-informed normally distributed emission probabilities that modeled a depleted state, a no-difference state, and an enriched state. The parameters and statepath were iteratively updated with Viterbi Training until the convergence of the log-likelihood of the Viterbi Encoded state path given

the parameters. The final state path that reached convergence was used as the final segmentation of the ChIP data into three domain types: depleted, no-difference, or enriched.

For mapping between euchromatin and heterochromatin (*Figure 3B*), the following limits of euchromatin were used:

chr = "2L" & start <21400000; chr = "2R" & start >5970000; chr = "3L" & start <22270000
chr = "3R" & start >4191000; chr == "X" & start <21400000

## ATAC-seq analysis

75 b.p. single-ended ATAC-seq reads are aligned and analyzed similarly as ChIP-seq data. Additionally, ATAC-seq peak calling across multiple samples is performed with the Genrich package ATAC-seq mode (0.6) (*Gaspar, 2021*).

## Heterochromatin analysis based on H3K9me3 ChIp-seq

To examine the heterochromatin distribution on the chromosome scale, processed H3K9me3 ChIp-seq data were further segmented into 250 kb bins using the bedtools multicov function (*Quinlan and Hall, 2010*). Reads were scaled by the sequencing depth. Corresponding input controls were used to eliminate backgrounds. Signals were then normalized with the euchromatin regions in between stages. The following limits of euchromatin at the 250 kb bin resolution (deduced from the nearby reported euchromatin limits) were used: chr2L, 0–21,500,000; chr2R, 5,500,000–25,286,936; chr3L, 0–23,000,000; chr3R, 4,250,000–32,079,331; chrX, 0–21,500,000. Processed data were visualized on the line charts.

To examine the heterochromatin distribution on repetitive sequences, reads were reprocessed with the similar pipeline as detailed above with a few alterations. Specifically, reads were remapped to a customized masked genome (repetitive sequences masked) with the addition of individual repetitive sequences as separated entries. Reads were scaled and normalized with the sequencing depth of euchromatin regions and corresponding input controls. Signals were quantified for individual repetitive sequences including transposons (TE) and satellites and visualized on the scatter plots.

## Acknowledgements

We thank Dr. Fredrick Tan and Allison Pinder for assistance with sequencing and genomic analysis. Mahmud Siddiqi provided valuable assistance with microscopy. We thank Drs. Rebecca Frederick and Lucy Morris for assistance in analyzing Janelia lines. We thank Gerry Rubin, Todd Laverty, and the Janelia Farm external investigator program for supporting the Spradling lab analysis of ovary expression using a sample of the Rubin enhancer collection. We thank current Spradling lab members for comments on the manuscript. Funding: Steve Z DeLuca was a postdoctoral fellow of the Helen Hay Whitney Foundation. Haolong Zhu and Liang-Yu Pang are/were graduate students in the Cell, Molecular and Developmental Biology Program at Johns Hopkins University, Department of Biology. Allan Spradling is an Investigator and John Urban is an Associate of the Howard Hughes Medical Institute.

## Additional information

### Funding

| Funder | Grant reference number | Author |
| --- | --- | --- |
| Howard Hughes Medical Institute | | John M Urban |
| Helen Hay Whitney Foundation | | Steven DeLuca |
| Johns Hopkins University | CMDB graduate program | Liang-Yu Pang Haolong Zhu |

The funders had no role in study design, data collection and interpretation, or the decision to submit the work for publication.

## Author contributions

Liang-Yu Pang, Data curation, Formal analysis, Validation, Investigation, Writing – original draft; Steven DeLuca, Conceptualization, Resources, Data curation, Formal analysis, Validation, Investigation, Writing – original draft, Writing – review and editing; Haolong Zhu, Data curation, Formal analysis, Investigation, Methodology, Writing – review and editing; John M Urban, Resources, Data curation, Software, Formal analysis, Investigation, Methodology, Writing – review and editing; Allan C Spradling, Conceptualization, Formal analysis, Supervision, Funding acquisition, Investigation, Writing – original draft, Project administration, Writing – review and editing

## Author ORCIDs

Haolong Zhu http://orcid.org/0000-0002-9056-0268
Allan C Spradling http://orcid.org/0000-0002-5251-1801

Reviewer #1 (Public Review): https://doi.org/10.7554/eLife.90509.2.sa1
Reviewer #2 (Public Review): https://doi.org/10.7554/eLife.90509.2.sa2

## Additional files

### Supplementary files

• Supplementary file 1. The Drosophila female germ cell cycle. (a) All processed data. (b) E(z)-dependent gene downregulation. (c) Mostly heterochromatic germline stem cell (GSC) genes with high H3K27ac. (d) Genes upregulated in young follicles relative to GSCs. (e) 41 membrane transporters upregulated more than 10-fold in young follicles.

• MDAR checklist

### Data availability

High throughput sequencing data and processed data is available under accession GSE145282 and GSE229943. Software code is available on GitHub (*DeLuca et al., 2020*; copy archived here).

The following dataset was generated:

| Author(s) | Year | Dataset title | Dataset URL | Database and Identifier |
|---|---|---|---|---|
| Pang L, Deluca SZ, Zhu H, Urban JM, Spradling AC | 2023 | Chromatin and gene expression changes during female *Drosophila* germline stem cell development | https://www.ncbi.nlm.nih.gov/geo/query/acc.cgi?acc=GSE229943 | NCBI Gene Expression Omnibus, GSE229943 |

The following previously published dataset was used:

| Author(s) | Year | Dataset title | Dataset URL | Database and Identifier |
|---|---|---|---|---|
| DeLuca S | 2020 | Differentiating *Drosophila* female germ cells initiate Polycomb silencing by regulating PRC2-interacting proteins | https://www.ncbi.nlm.nih.gov/geo/query/acc.cgi?acc=GSE145282 | NCBI Gene Expression Omnibus, GSE145282 |

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
