## [Editor Report · eLife assessment]

This **important** work significantly advances our comprehension of the molecular events occurring during germline stem cell differentiation in the *Drosophila melanogaster* ovary. The conclusions are strongly supported by **compelling** evidence, including rigorous data sets and complementary whole-genome analyses. As a result, this research holds substantial interest for developmental and stem cell biologists alike.

---

## [Referee Report · Reviewer #1 (Public Review)]

Summary:

The authors use a combination of ChIP-seq, RNA-seq and ATAC-seq on FACS-purified germ cells to understand the changes in transcription and chromatin landscape of germline stem cells (GSCs) and their progeny during adult oogenesis.

Strengths:

The major strengths of the paper include high quality -omics data, robust analyses of the data, and a well-written manuscript. The data strongly support the conclusions (1) that GSCs have more open chromatin than its differentiating daughter cells, (2) that H3K9me3 heterochromatin forms in 16-cell cyst stage and silences GSC-enriched genes, transposons, and testis-biased and somatic genes; (3) that GSC-enriched genes encoding cell cycle control, protein synthesis and signal transduction reside in clusters in autosomal pericentric regions; and (4) that there is a transcriptionally-driven metabolic reprogramming of nurse (germline) cells to aerobic glycolysis.

These data sets and analyses will have a high impact of the field of germline lifecycle (from GSC to primordial germ cells to GSCs again). The authors will make these data sets available through NCBI GEO and on Github. However, these are not incredibly user friendly and these data sets are extremely useful. I wonder if it is possible to incorporate these valuable data set into existing websites?

Weaknesses:

There are no obvious weaknesses.

---

## [Referee Report · Reviewer #2 (Public Review)]

Summary:

The preprint by Pang, Deluca, et al. investigates the molecular events occurring during germline stem cell (GSC) differentiation into an oocyte. The study highlights several critical observations:

1. Gene Expression and Chromatin State: GSCs exhibit an open chromatin state and express a large number of genes. However, during differentiation, the number of genes expressed decreases.

2. Gene Clustering and Chromatin Domains: Genes promoting GSC fate are found in clusters close to centric heterochromatin domains.

3. Epigenetic Marks: The transition from GSC fate to oocyte/nurse cell fate is marked by an increase in H3K27me3 and H3K9me3 on regions, including centric heterochromatin.

4. Metabolic Rewiring: Genes related to metabolism undergo changes during this fate transition, indicating metabolic rewiring.

Strengths:

The conclusions are strongly supported by a substantial amount of data. Multiple complementary methods are employed, such as increased H3K9me3 heterochromatin and reporter assays, to validate the increase of H3K9me3 during meiosis.

The wealth of data presented will be valuable to the scientific community, providing further insights into critical molecular events during GSC differentiation.

The study uncovers new biology, notably the proximity of stem cell genes to centric heterochromatin and its regulation.

Key observations include the low H3K9me3 levels on transposons in GSCs, which warrant further investigation.

Weaknesses:

To make the paper more accessible to a broader audience, the authors can use fewer jargon terms. In particular, the abbreviations used for staging can be confusing.

Some sections in the results contain extensive discussion that may be better suited for the discussion section. For example, see page 9.